# Monitoring Cochlear Nerve Action Potential for Hearing Preservation in Medium/Large Vestibular Schwannoma Surgery: Tips and Pitfalls

**DOI:** 10.3390/jcm12216906

**Published:** 2023-11-02

**Authors:** Baptiste Hochet, Hannah Daoudi, Etienne Lefevre, Yann Nguyen, Isabelle Bernat, Olivier Sterkers, Ghizlene Lahlou, Michel Kalamarides

**Affiliations:** 1Département d’Oto-Rhino-Laryngologie, Groupe Hospitalo-Universitaire Pitié-Salpêtrière, APHP, Sorbonne Université, 75013 Paris, Franceolivier.sterkers@aphp.fr (O.S.); ghizlene.lahlou@aphp.fr (G.L.); 2Technologies and Gene Therapy for Deafness, Institut de l’Audition/Institut Pasteur, 75012 Paris, France; 3Département de Neurochirurgie, Groupe Hospitalo-Universitaire Pitié-Salpêtrière, APHP, Sorbonne Université, 75013 Paris, France; etienne.lefevre@aphp.fr (E.L.);; 4Département de Neurophysiologie, Groupe Hospitalier Pitié-Salpétrière, APHP, Sorbonne Université, 75013 Paris, France; 5CRICM INSERM U1127 CNRS UMR 7225, Paris Brain Institute, Genetics and Development of Brain Tumors, 75013 Paris, France

**Keywords:** acoustic neuroma, CNAP, cochlear nerve, hearing, retrosigmoid approach

## Abstract

The diagnosis of large vestibular schwannomas (VS) with retained useful hearing has become increasingly common. Preservation of facial nerve (FN) function has improved using intraoperative EMG monitoring, hearing preservation remains challenging, with the recent use of cochlear nerve action potential (CNAP) monitoring. This prospective longitudinal series of VS with useful hearing operated on using a retrosigmoid approach included 37 patients with a mean largest extrameatal VS. diameter of 25 ± 8.7 mm (81% of Koos stage 4). CNAP was detected in 51% of patients, while auditory brainstem responses (ABR) were present in 22%. Patients were divided into two groups based on the initial intraoperative CNAP status, whether it was present or absent. FN function was preserved (grade I–II) in 95% of cases at 6 months. Serviceable hearing (class A + B) was preserved in 16% of the cases, while 27% retained hearing with intelligibility (class A–C). Hearing with intelligibility (class A–C) was preserved in 42% of cases when CNAP could be monitored in the early stages of VS resection versus 11% when it was initially absent. Changes in both the approach to the cochlear nerve and VS resection are mandatory in preserving CNAP and improve the rate of hearing preservation.

## 1. Introduction

When vestibular schwannoma (VS) tumors are too large for either a wait-and-scan strategy or radiosurgery; the main therapeutic option is tumor resection [1,2]. However, it must minimize sequelae on facial function and preserve hearing at a useful level if possible. The widespread use of MRI had increased the incidence of VS [3], and a growing portion of patients are diagnosed with large tumors when their hearing remains useful [4,5]. The challenge of today’s surgical management is to preserve this useful hearing in order to minimize the impact on postoperative quality of life [6]

Intraoperative electrophysiological facial nerve monitoring (IOFNM) is routinely performed to avoid the onset of a severe nerve conduction block, guiding the degree of VS resection to limit post-operative facial nerve palsy to 5% or less [4,7,8].

Synchronized auditory brainstem response (ABR) was the only positive predictive factor described in a recent study, with a five times greater chance of hearing preservation than desynchronized ABR [4]. To preserve hearing function, intraoperative ABR monitoring is routinely performed but with several limitations, especially for large VS. On one hand, preoperative ABR are desynchronized in many cases even though hearing is still useful, and on the other hand, useful real-time monitoring is not possible as a few minutes are required to obtain reliable waves. The modifications of ABR monitoring arise too late to change the ongoing surgical procedure, although it trains the surgeon to be less aggressive to the neural structures [9].

Cochlear nerve compound action potential (CNAP) monitoring is an intraoperative, continuous, near-real-time monitoring of cochlear nerve (CN) function through acoustic stimulation in the external ear canal. It is recorded using a specially designed CNAP electrode placed on the CN [10,11,12]. Its efficiency in hearing preservation has been reported for small VS removal (less than 15–20 mm extrameatal diameter) using a retrosigmoid (RS) approach [13], for which the CN is easily identified at the brainstem at the beginning of tumor dissection, with an 86% success rate for hearing preservation. Ishikawa et al. have recently reported hearing preservation in a single large VS (26 mm) case using CNAP monitoring, although the CN could not really be identified during surgery [14].

In this study, we discuss our preliminary experience with intraoperative CNAP monitoring during resection of medium/large VS. Tips and pitfalls in obtaining CNAP are reported.

## 2. Material and Methods

A prospective monocentric longitudinal study was conducted between September 2018 and July 2019. Inclusion criteria were surgery for medium/large VS (Koos 2 to 4) in patients with serviceable hearing (classes A and B) and useful hearing with intelligibility (class C) according to the American Academy of Otolaryngology-Head and Neck Surgery (AAO-HNS) classification [15]. Informed consent of patients was collected with approval from the Commission Nationale de l’Informatique et des Libertés (2211758). All data are reported according to the STROBE guidelines.

### 2.1. Preoperative Data

Tumors were staged according to both Koos classification [16] and the largest extrameatal diameter [5]. Hearing was assessed in a soundproof chamber in a quiet environment and included tonal audiometry and speech intelligibility with monosyllabic word list. The pure tone average was calculated as the mean of 500, 1000, 2000 and 3000 Hz thresholds and the speech discrimination score (SDS) was noted according to AAO-HNS recommendations. ABR were classified in 2 classes: synchronized (normal or delayed V wave) or desynchronized (no reproducible waves) [4]. Facial nerve function was reported according to the House-Brackmann grading system (H-B) [17].

### 2.2. Intraoperative CN Monitoring (IOCNM)

IOCNM was performed using Neuropack (Inomed, Emmendingen, Germany). Electrodes were positioned with one active electrode in front of the tragus of the operated ear (outside the surgical field), one electrode at the contralateral mastoid, one reference electrode at the vertex level, and one neutral electrode (ground) on the ipsilateral shoulder. Stimulation was performed using intra-ear headphones. CNAP monitoring did not extend surgical time.

CNAP data could be collected very rapidly (3–5 s). The stimulus used was rarefaction clicks at 95 dB HL and 11 Hz, and the amplitude and latency of N1 were obtained (Figure 1) [18,19]. Forty stimuli were given per acquisition, and a minimum of two acquisitions were necessary to verify the reproducibility of the plots obtained. The amplitude and latency of N1 were collected at the brainstem as soon as the CN was identified during VS dissection in the cerebellopontine-angle (CPA), then a continuous recording of the CNAP was pursued during VS resection. At the end of the procedure, the CNAP was recorded at the brainstem, the internal auditory meatus (IAM) and the fundus. In the case of a decrease in wave amplitude by more than 50% or its complete loss during VS resection, the surgical procedure was transiently interrupted.

ABR were measured from the beginning of the intervention, and continuously during tumor dissection if present. Per acquisition, 1000 stimuli were given (90 s), and a minimum of two acquisitions were necessary to verify good reproducibility. The presence, amplitude and latency of wave V were considered. In the case of an increase in wave V latency by more than 0.4 ms or wave loss, the surgical procedure was transiently interrupted.

### 2.3. Intraoperative Facial Nersurfaceve Monitoring (IOFNM)

IOFNM was performed using an NIM 3 response device (Medtronic, Jacksonville, FL, USA). If the amplitude response after a supramaximal stimulation (2 mA) decreased by more than 40% or if a very significant adhesion was present, dissection of the facial nerve from the tumor was interrupted and the tumor resected around it leading to different degrees of resection [3,12].

### 2.4. Surgical Technique

After having opened the dura and retracted the cerebellum through RS approach it was, in most cases of large VS, difficult or even impossible to identify the facial and cochlear nerves at the brainstem, so blind mapping of the posterior VS surface was performed using both facial nerve monitoring and the ball-electrode for CNAP monitoring from the Inomed device (Figure 2A). The first step was to debulk as far as possible the posterior VS extension. Then, its inferior pole was carefully resected to expose the cochlear and facial nerves (Figure 2B). Direct recording of the CNAP could be reported by placing the ball-electrode on the nerve, hearing being controlled since the beginning of the surgical procedure by ABR monitoring when present. Thus, tumor removal was performed progressively, and transiently interrupted if CNAP monitoring issued an alert.

When the facial nerve was displaced superiorly, this surgical strategy was not possible as the cochlear and facial nerves are diametrically separated by a piece of tumor. In those cases, VS resection could be interrupted to preserve normal postoperative facial function, leaving a tumor remnant of greater importance along the facial nerve. Then, the IAC was drilled to expose and remove the intrameatal part of the VS, which is relatively straightforward depending on the extension at the fundus and the texture of the tumor. At this time, the ball-electrode was moved to ensure the integrity of the CN and compare the CNAP responses between the surface on the VS, the fundus, and the brainstem (Figure 2C). As for the preservation of the facial nerve function, near/sub-total VS resection has been performed in some cases to maintain the integrity of the cochlear nerve predicted by CNAP monitoring (Figure 2D).

### 2.5. Postoperative Data

Follow-up was 6 months with MRI, facial nerve and hearing evaluations. Degree of VS resection was assessed at the end of VS removal by the surgeon’s evaluation as gross-total resection (GTR) or not, and then classified according to the early postoperative MRI scan as: total resection if there was no tumor residue confirmed by MRI; Near-total resection (NTR) if the small tumor residue was not contrast-enhanced; Sub-total resection (STR) if the tumor residue was smaller than 0.5 cm^3^; Partial resection (PR), mostly preoperatively planned, if the tumor residue was greater than 0.5 cm^3^ [20].

### 2.6. Statistical Analysis

All data are expressed as mean ± standard deviation (SD), [min-max], or number (percentage). Statistical analysis was performed using Prism (version 8.4.0, GraphPad Software, San Diego, CA, USA). Mann-Withney, *t*-test, Chi-squared and Fisher’s tests were used. A *p*-value < 0.05 was considered to be statistically significant.

## 3. Results

In total, 37 patients were included (25F/12M) in this study with an average age of 43 ± 13.1 years (19–71). Two patients had neurofibromatosis type 2 and two others had been treated previously by radiosurgery. The largest mean diameter of extrameatal VS was 25 mm ± 8.7 mm (12–46) with 34 (92%) Koos 3 and 4 VS and 32 (86%) Sterkers 3 and 4 VS (Table 1). The tumor was cystic in 7 cases (19%). IAM invasion was complete, partial or absent in 43%, 49%, and 8%, respectively.

Preoperative PTA was 32 ± 18.7 dB (3–70) with an average SDS of 90 ± 16% (50–100). Hearing was classified as class A for 20 (54%) patients, class B for 11 (30%) patients and class C for 6 (16%) patients. Eight patients (22%) had preoperative synchronized ABR. Facial nerve function was normal for all patients. VS resection was performed because of brainstem compression symptoms (*n* = 28; 76%), rapid tumor growth of >2 mm/year (n = 6; 16%), or slow tumor growth of ≤ 2 mm/year with recent trigeminal neuralgia (n = 3; 8%).

### 3.1. CNAP

The patients were divided into two groups: CNAP present (Group 1; *n* = 19, 51%) or absent (Group 2; *n* = 18, 49%) at the brainstem after CN identification (Figure 3). The two groups had similar preoperative characteristics (Table 1).

In Group 1 (n = 19), once CNAP was recorded, average values of N1 latency and amplitude were 4 ± 0.8 ms (2–6) and 12 ± 18 µV (2.1–5.5), respectively. CNAP at the brainstem was still present at the end of VS dissection in the CPA for 12 (63%) patients, and after IAM drilling and intrameatal tumor resection for 8 (42%) patients (Figure 3). The surgical procedure was transiently interrupted on average three times per intervention, for an average duration of 45 s each.

In Group 2 (n = 18), CNAP was not recorded for a number of presumed causes listed by order of frequency: CN unidentified or interrupted before recording in 8 cases (44%), unknown reasons with no recording at the fundus of the internal auditory canal (IAC) with intact CN (probably due to technical failure) in 4 (22%), presumed vascular damage as CNAP was not found from the brainstem to the IAC fundus and associated with complete hearing loss in 3 (17%), no CNAP together with ABR loss during the surgical approach before CN identification in 2 (11%), and technical problems as ABR were present, CN intact, and hearing unchanged in the last case.

### 3.2. ABR

Among the 8 (22%) patients with preoperative synchronized ABR, 5 had CNAP at the brainstem at the time of CN identification (Figure 3). At the end of surgery, only 3 patients (8%) still had synchronized ABR and CNAP at the brainstem, for whom a serviceable hearing was preserved post-operatively. The 3 others lost ABR before CN identification and no CNAP were recorded. patients without CNAP had lost ABR before CN identification.

### 3.3. Hearing Preservation

Hearing with intelligibility (hearing class A, B and C) was retained for 10 (27%) patients and serviceable hearing (hearing class A and B) for 6 (16%) patients (Figure 3 and Table 2).

When hearing with intelligibility was preserved, CNAP at the brainstem was recorded at the end of surgery for 7 of these 10 (70%) patients (See example Figure 4). For the three others, CNAP could not be monitored at the brainstem but at the fundus even under steroid therapy which may be accounted for by the onset of a nerve conduction blockage along the CN.

In the case of postoperative anacusis (n = 27), CNAP was no longer present at the brainstem at the end of surgery in all cases but one, who experienced a sudden hearing loss 4 h after surgery, presumably due to cochlear ischemia (Figure 3). In Group 1, postoperative hearing with intelligibility (hearing class A, B and C) and serviceable hearing (hearing class A and B) were retained for 8 (42%) and 4 (21%) patients, respectively (Table 2). In cases of hearing loss, CNAP losses were observed after CPA, IAC, or porus dissection in 55%, 27% or 18% cases, respectively. In 16 (84%) patients of Group 1, ABR at the end of surgery were desynchronized.

Suspected causes of hearing loss during tumor removal in Group 1 (11 patients) were: microvascular damage (CNAP absent at the brainstem and at the fundus, Figure 5) in 3 cases; anatomic lesion to the CN during dissection in the CPA in 2 others (CNAP present at the fundus); nerve conduction block (CNAP absent at the brainstem but present at the IAC fundus and intact CN, Figure 6) in 1 other; delayed hearing loss in another; unidentified (CNAP loss during CPA dissection, no IAC background test, intact CN) in 4 cases.

In Group 2 (18 patients) with absence of CNAP during surgery, serviceable hearing was retained in only 11% (2/18) of cases and there were no additional cases with class C hearing.

Postoperative hearing with intelligibility was achieved less frequently in Group 2 than in Group 1, although not significantly different (*p* = 0.06, Fisher’s test).

Neither N1 latencies and amplitudes at the brainstem at the beginning of VS dissection nor CNAP amplitude decrease between the IAC fundus and the brainstem at the end of the procedure were correlated with preservation of hearing with intelligibility (Table 3).

### 3.4. Postoperative Facial Nerve Function

Facial nerve function was good (grade I and II) for 30 (81%) patients at discharge, and for 35 (95%) patients (28 grade I, 7 grade II) 6 months after surgery. Only one grade III and one grade VI were still present at 6 months. Postoperative facial nerve function was not different between the two groups (Table 2).

### 3.5. Degree of Resection

VS resections were classified as TR, NTR, STR, and PR for 21 (57%), 7 (19%), 5 (13%), and 4 (11%) patients, respectively. No statistically significant difference was found between the two groups (*p* = 0.90, Table 2).

## 4. Discussion and Conclusions

In the last few years, our neuro-otological team has made a paradigm shift in the surgical management of medium/large VS toward their removal through a RS approach, instead of a translabyrinthine (TL) approach, in an attempt to preserve hearing together with a stable high rate of postoperative normal facial nerve function [4]. It is not a new concept, especially in the neurosurgical community, to try to preserve hearing for large VS using sub-occipital or RS approaches [21,22,23], although some of them were more prone to use the TL approach for large tumors whatever the preoperative hearing status [24].

In many institutions, small VS are resected routinely under intraoperative ABR monitoring, but the effects of surgical manipulations on hearing are observed with a delay, which precludes any reversible maneuvers. At variance, as CNAP is rapidly recorded (every 3–5 s) providing immediately reliable information on CN function, tumor dissection can be adapted to these real-time changes.

In the present study, reporting intraoperative auditory monitoring during medium/large VS (mean extrameatal diameter of 25 mm) resections, ABR and CNAP were recorded in 24% and 51% of cases, respectively, which compares favorably to the seminal paper by Yamakami et al. [18] focused on small tumors (mean extrameatal diameter of 12 mm), with recording of ABR and CNAP in 34% and 66% of cases, respectively.

The reasons why hearing can be lost during tumor resection from the CN differ from those for the facial nerve. Cochlear monitoring has changed and improved gestures and surgical strategy for separation of the CN from the tumor (Figure 2). We performed systematic CN mapping of the posterior face of the tumor before any debulking, as was already performed for the facial nerve (Figure 2A). This allowed us to detect the non-visible root of the CN. Moreover, it is necessary to limit the dissection time on the CN by the widest debulking of the superior and posterior part of the VS, distant from the nerve. The CNAP recording ball can be moved for dynamic mapping, before seeing the CN on the inferior surface of the partially resected vs. (Figure 2B). Then, tumor dissection along the CN in the CPA is performed under continuous CNAP monitoring. It is sometimes difficult to stabilize the recording ball on the CN, particularly when there is almost no tumor in contact, and its replacement could make the interpretation of monitoring more difficult justifying the permanent presence of a collaborator during the surgical procedure.

CNAP monitoring is of primary importance during resection of the residual intracanalar tumor. After drilling of the porus, the ball is placed on the CN so that the residual tumor can be removed in a single piece under CNAP monitoring after resection of the vestibular nerves and separation from both cochlear and facial nerves (Figure 2C,D).

A particular cause of hearing loss is of vascular origin, accounting for at least 20% of hearing loss in our study, either by coagulation or spasm of the internal auditory (labyrinthine) artery. Accordingly, we observed that, after coagulating this artery by necessity to stop its bleeding in one case, the CNAP signal suddenly disappeared 15 min later at the fundus and did not recover (Figure 5). Coagulation on the tumor surface must be reduced as much as possible before debulking, although we did not experience such an adverse effect on intraoperative facial nerve monitoring (Figure 6). The addition of papaverine-soaked cottons for the dissection of the CN should be helpful to prevent any vascular damage. As reported recently, infrared video-angiographic assessment of the CN vasculature during VS surgery may be an attractive tool, particularly when the tumor is separated from the nerve [25].

Finally, in case of useful preoperative hearing, a near-/subtotal VS resection through a retrosigmoid approach could be guided by CNAP monitoring as is already performed using IOFNM to preserve the hearing and facial functions with the minimal tumor residue [8]. In case of total postoperative hearing loss with a preserved CN, cochlear implantation can be discussed, especially if the patient has disabling tinnitus or bilateral profound hearing loss, yet quite rare for sporadic VS [26]. Promontory stimulation test has been commonly described as a prognostic tool for cochlear nerve function, used with electrically evoked auditory brainstem response (eABR): it has a high sensitivity for CI use and auditory performance, with 87% of patients with positive eABR having a speech intelligibility with their CI [27]. This strategy is also used during simultaneous translabyrinthine VS resection with ipsilateral cochlear implantation to assess that the integrity of the CN [28] and could be proposed in case of preoperative non-useful hearing, with the aim of improving hearing in noise and tinnitus after the surgery [29].

The main limitation of this study is its low power, due to the small sample studied.

Despite the lack of better results for hearing preservation, our early experience of CNAP monitoring during surgery for the largest VSs through a RS approach has convinced us that this innovation is a major advance in the field of VS surgery allowing atraumatic dissection of the CN with a significant increase in hearing preservation when CNAP are still present. For these large VSs, care should be taken to maintain CNAP during the approach to the CN, and then to guide the degree of VS resection under the control of both facial and cochlear nerve monitoring.

## Figures and Tables

**Figure 1 jcm-12-06906-f001:**
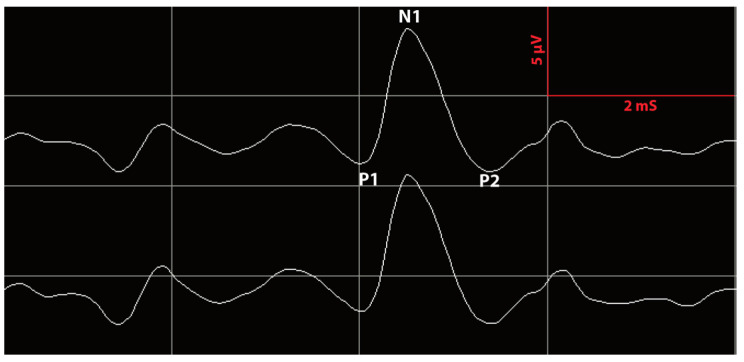
Screenshot of cochlear nerve action potential (CNAP) monitoring. Amplitudes in microvolts and latencies in milliseconds.

**Figure 2 jcm-12-06906-f002:**
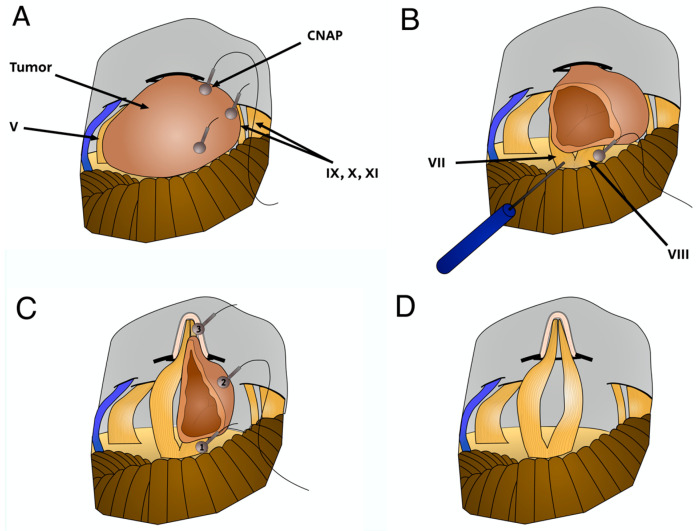
Surgical strategy for the electrode-ball placement during VS removal under CNAP monitoring. Retrosigmoid approach, surgical view. Blind mapping of the posterior VS surface (**A**). The electrode-ball was placed on the cochlear nerve (CN) after debulking of the superior and inferior tumor poles (**B**). After drilling the IAC posterior wall, the electrode-ball was moved to ensure the integrity of the CN and compare the CNAP responses between the surface on the brainstem (1), the VS (2), the fundus (3) (**C**). Surgical field view at the end of a GTR (**D**). V: trigeminal nerve, IX, X, XI: cranial caudal nerves.

**Figure 3 jcm-12-06906-f003:**
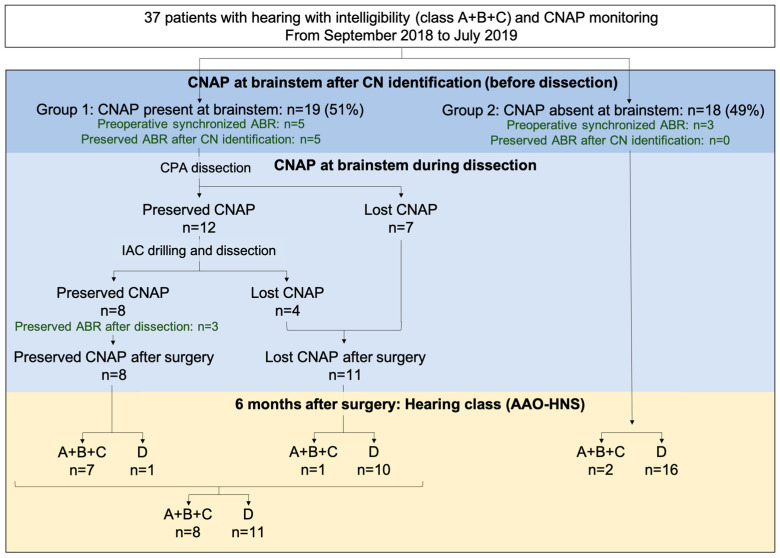
Flow chart of per-operative CNAP monitoring and hearing preservation results. ABR: auditory brainstem response; AAO-HNS: American Academy of Otolaryngology-Head and Neck Surgery; CNAP: cochlear nerve action potential; IAC: internal auditory canal.

**Figure 4 jcm-12-06906-f004:**
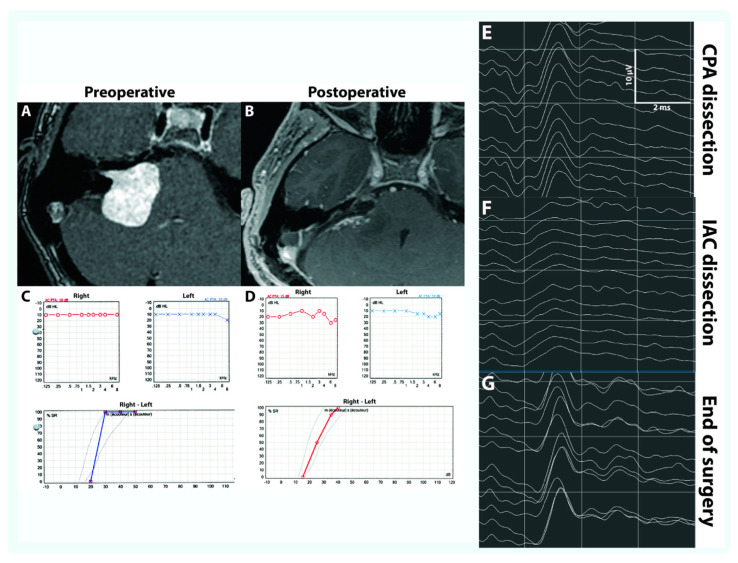
A 39–year-old woman presented with stage 4 VS of 34 mm diameter (**A**) with total invasion of the IAC (grade A) and hearing class A (**C**). There were no preoperative ABR. Resection of the VS was performed using an RS approach. After debulking and some tumor removal, there was a good CNAP amplitude of 6.8 µV at the time of cochlear nerve identification. There was no change in CNAP after tumor removal from the CPA (**E**). During cochlear nerve dissection from the tumor in the IAC, CNAP amplitude decreased by 50% (**F**). Surgery was transiently interrupted then resumed after a few minutes. At the end of surgery, there was no change in CNAP compared to the start of surgery (**G**). Resection of the tumor was total (**B**) and postoperative facial function was grade 1. Postoperative audiometry was comparable to preoperative measurements with hearing class A at 6 weeks (**D**).

**Figure 5 jcm-12-06906-f005:**
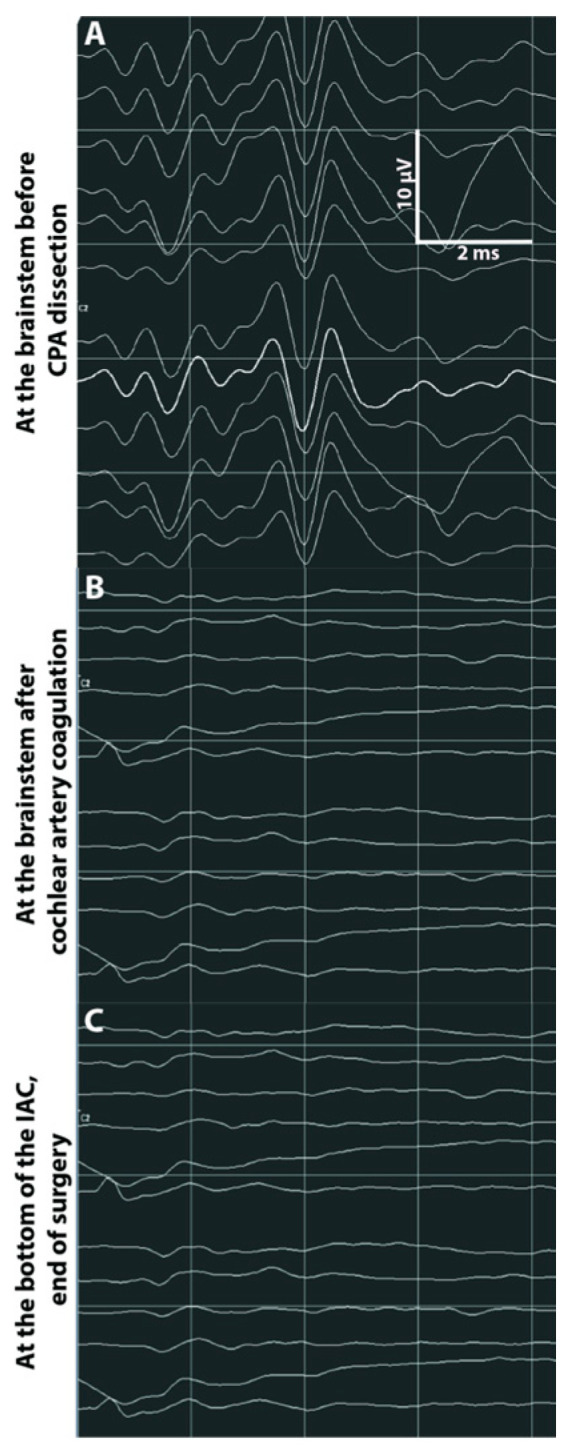
A 55-year-old woman presented with very rapidly growing stage 3 VS of 17 mm diameter with partial IAC invasion (grade B) and hearing class B. Resection of the VS by a RS approach was decided. After debulking and some tumor removal, there was a good CNAP amplitude (5.1 µV) at the time of cochlear nerve identification (**A**). Because of bleeding, there was a progressive loss of ABR and then CNAP 20 min after coagulation of the internal auditory artery during tumor dissection in the CPA (**B**). CNAP was absent at IAC fundus (**C**). Resection of the tumor was total and immediate postoperative facial function was grade 3. Postoperative audiometry confirmed anacusis (hearing class D).

**Figure 6 jcm-12-06906-f006:**
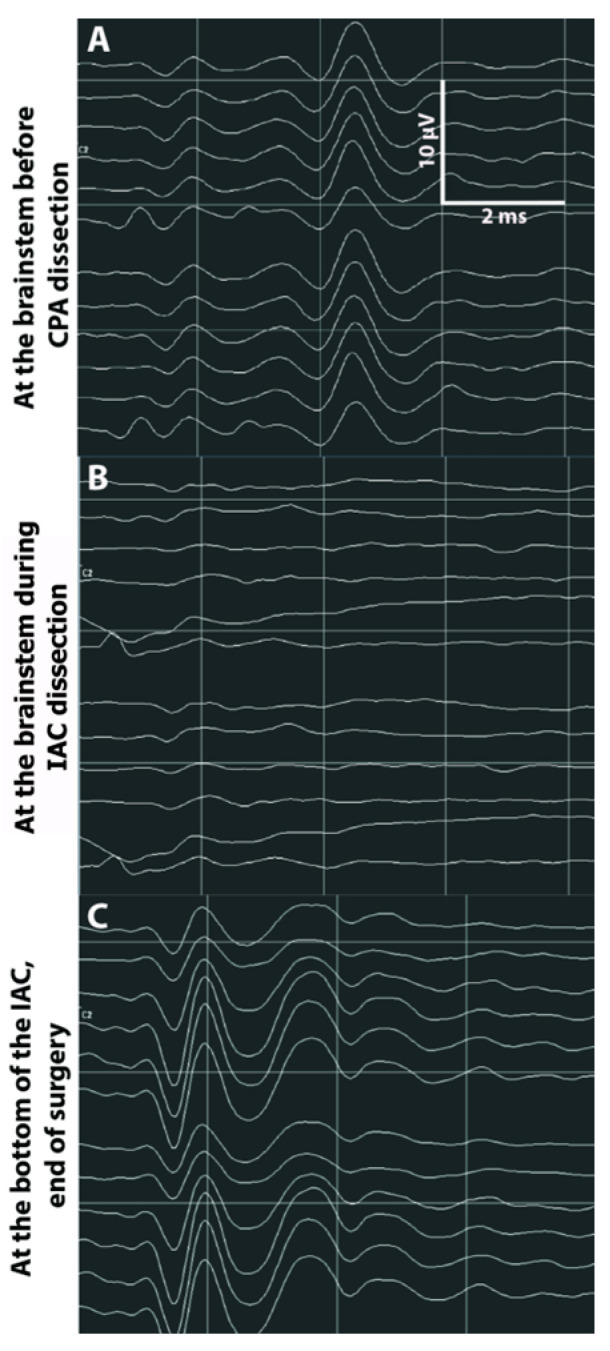
A 49-year-old woman presented with stage 4 VS of 25 mm diameter with partial IAC invasion (grade B) and preoperative hearing class A. Preoperative ABR were synchronized. It was decided to operate through a RS approach. After debulking and some tumor removal, there was a good CNAP amplitude (7.8 µV) at the time of cochlear nerve identification (**A**). CNAP at the brainstem was stable at the end of tumor dissection in the CPA. There was a sudden loss of CNAP at the brainstem during dissection of the cochlear nerve in the IAC (**B**). Despite stopping the surgical procedure for several minutes, there was no recovery of CNAP. At the end of the procedure, CNAP was absent from the brainstem but present at the IAC fundus indicating nerve conduction block (**C**). Resection of the tumor was total and postoperative facial function was grade 1. Postoperative audiometry confirmed anacusis despite corticosteroid treatment (hearing class D).

**Table 1 jcm-12-06906-t001:** Preoperative radiological and audiological characteristics of vestibular schwannoma tumors for the 37 patients in this study.

	Group 1 (*n* = 19)		Group 2 (*n* = 18)		*p*-Value
KOOS stage, n (%)					
1	0	(0)	0	(0)	0.1 ^¥^
2	3	(16)	0	(0)	
3	2	(10)	2	(11)	
4	14	(74)	16	(89)	
Sterkers’ stage, n (%)					
1	0	(0)	0	(0)	0.6 ^¥^
2	4	(21)	1	(6)	
3	10	(53)	13	(72)	
4	5	(26)	4	(22)	
Largest diameter in CPA [mm, mean ± SD (range)]	23 ± 8.5 (12–40)		27 ± 7.7 (20–46)		0.09 ^∞^
Mean tonal threshold [dB, mean ± SD (range)]	28 ± 18.1 (2–65)		35 ± 18.9 (6–64)		0.9 ^‡^
Speech discrimination [%, mean ± SD (range)]	89 ± 17.2 (50–100)		93 ± 14.1 (50–100)		0.4 ^∞^
Hearing class (AAO-HNS)					
A	11	(58)	9	(50)	0.7 ^¥^
B	5	(26)	6	(33)	
C	3	(16)	3	(17)	
ABR					
Synchronized	6	(32)	2	(11)	0.2 ^¶^
Non-synchronized	13	(68)	16	(89)	

ABR: Auditory Brainstem Responses; AAO-HNS: American Association of Otolaryngology and Head and Neck Surgery; CPA: cerebellopontine angle Sterkers’ stage according to the largest diameter in the cerebellopontine angle (CPA): stage 1 for intracanalar tumor; stage 2 if the largest diameter in CPA is <16 mm; stage 3 if the largest diameter in CPA is between 16 and 29 mm; stage 4 if the largest diameter in CPA is ≥30 mm. Data are mean ± SD (range) or n (%); ^¥^: Chi-squared test; ^∞^: Mann–Whitney test; ^‡^: Student *t*-test; ^¶^: Fisher test.

**Table 2 jcm-12-06906-t002:** Postoperative features depending on the presence (Group 1) or the absence (Group 2) of CNAP at the brainstem at the beginning of surgery.

	Group 1 (n = 19)		Group 2 (n = 18)		*p*-Value
**Hearing Class (AAO-HNS)**					
A	3	(16)	1	(6)	0.1 ^¥^
B	1	(5)	1	(6)	
C	4	(21)	0	(0)	
D	11	(58)	16	(88)	
Mean tonal threshold [dB, mean ± SD (range)]	88 ± 41.9 (15–120)		104 ± 37.7 (13–120)		0.2 ^∞^
Speech discrimination score [%, mean ± SD (range)]	32 ± 41.9 (0–100)		11 ± 32.3 (0–100)		0.06 ^∞^
Facial nerve function at 6 months (HB)					
I	13	(68)	15	(83)	>0.99 ^‡^
II	5	(26)	2	(11)	
III	0	(0)	1	(6)	
IV	0	(0)	0	(0)	
V	0	(0)	0	(0)	
VI	1	(5)	0	(0)	
Degree of resection					
TR	13	(68)	8	(44)	0.9 ^¥^
NTR	3	(16)	4	(22)	
STR	2	(11)	3	(17)	
PR	1	(5)	3	(17)	

AAO-HNS: American Association of Otolaryngology and Head and Neck Surgery; CNAP: cochlear nerve action potential; HB: House-Brackmann; TR: total resection; NTR: Near-total resection; PR: Partial resection; STR: Sub-total resection; Data are mean ± SD (range) or n (%); ^¥^: Chi-squared test; ^∞^: Mann–Whitney test; ^‡^: Fisher test.

**Table 3 jcm-12-06906-t003:** Analysis of CNAP recording in Group 1 depending on the postoperative auditory outcome.

	Postoperative Class A, B and C	Postoperative Class D	*p*-Value
N1 latency at the beginning of surgery (ms)	18 ± 27.5 (5–80)	7 ± 4.8 (1.4–14)	0.3
N1 amplitude at the beginning of surgery (µV)	4 ± 0.9 (3.4–5.5)	3 ± 0.8 (2.1–4.6)	0.16
CNAP amplitude decrease (%)	72 ± 25.7 (40–100)	93 ± 15.7 (65–100)	0.1

ms: milliseconds; µV: millivolts; CNAP: cochlear nerve action potential. Hearing is classified according to the American Association of Otolaryngology and Head and Neck Surgery recommendations. Data are mean ± SD (range); statistical analyses were performed using the Mann–Whitney test.

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
