# Peer review of "Monitoring Cochlear Nerve Action Potential for Hearing Preservation in Medium/Large Vestibular Schwannoma Surgery: Tips and Pitfalls"

_jcm, 2023, doi:10.3390/jcm12216906_

Round 1

Reviewer 1 Report

Comments and Suggestions for Authors

Dear Editor,

I reviewed the article entitled “Monitoring Cochlear Nerve Action Potential for Hearing Preservation in Medium/Large Vestibular Schwannoma Surgery: Tips and Pitfalls” by Hochetet al discussing the role of cochlear nerve action potential as a potential tool to manage the surgery for medium/large vestibular schwannomas.

The article is well written, consistent concerning the data and it may be useful for ENT doctors and the whole scientific community.

It provides a new approach to the live monitoring of this kind of surgery and the data seems to be on a significant cohort of patients.

Consequently, the article seems to be suitable for publication.

Reviewer 2 Report

Comments and Suggestions for Authors

This study summarized the application of cochlear nerve action potential monitoring during the surgery of vestibular schwannoma and found that the monitoring may be useful for the preservation of hearing. The number of cases in this study is not large, but the contents may be useful for the clinicians.

Was the origin of vestibular schwannoma superior or inferior? Some previous articles have the information of the origin, which might be useful for the readers.

Line 19

… is increasing

Previous literatures should be cited in the main text.

Line 73

And.

And what?

Line 85

95 dB

dB nHL or dB SPL?

Line 168 Table 1, Line 225 Table 2

The line beneath “KOOS stage” and “Hearing class” end in the middle of the table. Is the length of this line correct?

Line 188

Technical failure?

“?” should not be used in the article.

Reviewer 3 Report

Comments and Suggestions for Authors

Thank you for inviting me to review this submission about cochlear nerve monitoring for vestibular schwannoma resection. Multiple approaches including watch and wait for these tumors have been described. In order to advance in the field, the strategy to do maximal safe resections in Neuro-oncology is paramount. This technique confers a new modality for functional preservation. I'd suggest improving a few grammatical and syntax errors throughout the manuscript. The grammar of the introduction must be improved. Otherwise, the surgical and monitoring techniques are well described, as well as the results. The discussion is sufficient and the conclusions are appropriate for the whole manuscript. This topic is relevant, and novel, and would add sufficient archivable value to the field. 

Comments on the Quality of English Language

Please revise the comments above. Revise the abstract as well as the introduction to improve the reading of the manuscript.

Reviewer 4 Report

Comments and Suggestions for Authors

Dear Authors, I read with great interest your valuable manuscript regarding management of VS and impact of intraoperative CNAP. Just some minor revisions: are you sure about consistency of citation 3 at line 42 and later? may you better specify the synchronized and desynchronized classification of preoperative ABR? Moreover, may you provide surgical time in both groups? The introduction and discussion section may benefit from a more comprensive literature evaluation. Consider also other citation from this special issue with the goal to reach 30 references. TL approach, cochlear nerve preservation and simultaneous CI is another emerging strategy you may discuss. 
